# Fabrication and Optimization of Nafion as a Protective Membrane for TiN-Based pH Sensors

**DOI:** 10.3390/s23042331

**Published:** 2023-02-20

**Authors:** Shimrith Paul Shylendra, Magdalena Wajrak, Kamal Alameh

**Affiliations:** School of Science, Edith Cowan University, Joondalup, WA 6027, Australia

**Keywords:** pH sensing, Nafion, titanium nitride, redox effects, outer membrane, medical applications

## Abstract

In this study, a solid-state modified pH sensor with RF magnetron sputtering technology was developed. The sensor consists of an active electrode consisting of a titanium nitride (TiN) film with a protective membrane of Nafion and a reference glass electrode of Ag/AgCl. The sensitivity of the pH sensor was investigated. Results show a sensor with excellent characteristics: sensitivity of 58.6 mV/pH for pH values from 2 to 12, very short response time of approximately 12 s in neutral pH solutions, and stability of less than 0.9 mV in 10 min duration. Further improvement in the performance of the TiN sensor was studied by application of a Nafion protective membrane. Nafion improves the sensor sensitivity close to Nernstian by maintaining a linear response. This paves the way to implement TiN with Nafion protection to block any interference species during real time applications in biosensing and medical diagnostic pH sensors.

## 1. Introduction

pH is an important parameter in a wide range of industrial and medical applications. Several recent studies have concentrated on the formation of novel solid-state pH sensors and accurate pH measurement techniques, particularly with the development of micro/nanotechnology and innovation [1,2]. A common way to measure pH is with a potentiometric sensor with a glass electrode. The first report on the use of a glass electrode as a pH sensor was by Max Cremer in 1906, where it was shown that an electrical potential across a glass membrane is proportional to the pH difference across that membrane. Since then, glass electrodes are routinely used in variety of situations where accurate and reliable pH measurement is required. Glass electrodes these days are can identify and quantify H+ ions in many systems and a variety of samples with high precision [3].

However, despite their accuracy, great sensitivity and good reliability, pH glass electrodes are too fragile and bulky in size for them to be used as miniaturized sensors for in vivo applications [2]. In addition, glass pH electrodes decompose over time leading to a decrease in pH sensitivity [2,4] and the deposits on the electrode membrane can affect pH measurements. It is also very important to always store the pH glass electrodes in an aqueous solution, most usually in 3 molL^−1^ KCl buffer solution when not in use and perform regular maintenance [5].

Solid-state potentiometric pH sensors are composed mainly of metal oxides and have several benefits over glass electrodes, such as variable shapes, the possibility of miniaturization, high selectivity [6], low manufacturing cost with high production rate, and possibility of integration of the sensor with real-time microelectronic monitoring components. Among iron oxides, RuO_2_ is highly prized for its attractive properties, such as high sensitivity [7], selectivity [8], good reaction even in the presence of highly oxidizing and reducing species [9], and stable performance in various samples [10]. However, there are still problems with the RuO_2_ solid-state metal oxide electrode regarding pH measurements in complex biological, environmental, and industrial media due to redox interference. For example, proteins and other macromolecules adsorb in the electrode area can also interfere with electrode function; this limits the use of RuO_2_ sensors in the measurement of pH in food matrixes (e.g., meat, fish, dairy products) and complex samples (wine and orange juice). To address this issue, the literature reports three main ways to block the electron movement between the metal-oxide sensor and solution being measured while maintaining proton conduction in the electrode area. The first method is a single point measurement protocol with an accuracy of ±0.2 pH obtained. The second method involves a matrix-compliant measurement protocol, which shows improved accuracy of ±0.1 pH compared to commercial pH sensors. The third method by Lonsdale et al. [11] uses sputter-deposit of a thin layer of Ta_2_O_5_ and drop-cast Nafion to reduce redox interference. Unfortunately, these methods are limited by their low accuracy, requirement for highly skilled operators [12] and inability to apply to more problematic matrices, such as wine and orange juice. Aimed at expanding the application area of solid-state sensors and developing and validating additional sample-standard compounds, this work investigated another type of solid-state sensors employing active electrodes based on titanium nitride (TiN) combined with Nafion films.

A TiN-based pH electrode has been previously investigated [13], showing efficacy in more problematic matrices containing redox species. However, the potential shift due to the presence of redox species for the TiN electrode was up to 30 mV. This shift is not acceptable and, therefore, further work was needed to minimize the redox problems of this sensor. To reduce this potential shift and make the sensor work more efficiently in challenging matrices, such as biological media, a Nafion film was applied to the TiN sensor. Nafion is an inactive substance that inhibits macromolecules, but at the same time facilitates the electron transfer between the sensor and the media [14,15]. Nafion consists of perfluorinated ionomer, which is a proton-conducting polymer. The structure of the perfluoro vinyl ether composite groups is terminated by sulfonate groups in the backbone of tetrafluoroethylene (PTFE), as shown in Figure 1, which enables Nafion to offer unique properties, such as high proton exchange, chemistry, and temperature stability [16,17].

Nafion’s strong ionic conductivity, cation selectivity, chemical inertness, and thermal stability make it a well-known ionic polymer [18]. Significant interest in the special features of Nafion has been shown in numerous research fields and applications, such as fuel cells and batteries [19], chemical sensors [20], biosensors [21,22,23], and medical diagnosis technologies [24,25]. Nafion has received a lot of attention recently in the context of electrochemical sensors for biomedical and environmental applications as a membrane or electrode modification [26,27,28]. The chemical structure of Nafion allows it to behave as a semipermeable membrane, where only protons can reach the electrode and all other types of ions are blocked. However, the Nafion protective layer embedded in previously studied metal oxide electrodes showed significant limitations in sensory use, as it significantly increased the reaction time, by up to 20 min in neutral pH environments and on average by 3 to 5 min in acidic and basic conditions [29]. The Grotthuss explanation [30], proposes that structural diffusion is responsible for the proton conductivity of fully hydrated Nafion. By rotating and reorienting water molecules, protons travel across the hydrogen bond within the water 3D network inside the polymer membrane [31]. According to Gierke’s model [32], hydrophilic sulfone groups coupled into channels keep water molecules in polymer clusters, allowing protons to flow continuously across the polymer membrane. When the SO_3_-H connection is broken, protons dissociate, which is the first step in the transfer of protons. Dissociated protons subsequently combine with water to generate the hydronium (H_3_O^+^), Eigen (H_9_O_4_^+^), and Zundel (H_5_O_2_^+^) ions, which move through the water network. This behavior of Nafion is due to its cationic nature, as demonstrated by Kinlen et al. [33], in which Nafion slightly inhibited the redox sensation of the IrO_2_ pH sensor at the expense of the sub-Nernstian slope sensor. Nafion has also been used by other research groups [34,35]; however, in these cases, Nafion was used to improve iron-oxide sensitivity by preventing its degradation or degradation. Manjakkal et al. [36] have reported another way to eliminate redox interference by placing a thin (76 nm) Ta_2_O_5_ layer on the sensitive electrode of the IrO_2_ pH sensor. Pre-concentrating target analytes and minimizing interference from anionic species have both been accomplished using Nafion membranes [37,38,39,40]. The membrane adheres well to most electrode surfaces and can prevent fouling and surface deterioration [41]. Anodic stripping voltammetry (ASV) and a Nafion-modified bismuth film sensor were used by Akl et al. to determine the levels of Pb (II) and Cd (II) in lake water in the presence of surfactants [42]. Nafion solution was drop cast onto the working electrode to create a 0.4 m thick membrane, with the detection threshold being approximately 0.5 g/L. In this work, it was discovered that the Nafion membrane’s thickness significantly affected both the sensor’s sensitivity and resistance to surfactants. Those studies confirmed that Nafion remains an effective redox inhibitor, and as a result, in this work, Nafion was applied to TiN as it is more cost-effective, and durable compared to metal-oxides. Membrane deposition and treatment parameters, such as deposition process and annealing/boiling temperature, affect the proton conductivity of Nafion. The Nafion membrane was deposited on the electrode surface by drop-casting a commercially available solution and sensing mechanism of the developed layers was reported previously [43]. However, methods presented in various research papers varied in the quantity of Nafion layers and in the method of drying the Nafion. There is also a lack of evidence in the literature on how TiN with Nafion layer works to prevent redox species in the sample solution. For that reason, this work aims to investigate Nafion coating on the performance of TiN electrodes in the form of RFMS using metal-oxide pH electrodes as counterparts. Cationic Nafion research was performed in this study to determine the effect of Nafion during the reaction of the TiN sensor and its pH sensitivity with the aim of using this sensor in industrial, environmental, and medical applications.

## 2. Methodology

### 2.1. Production of TiN Electrodes and Nafion Deposition

Active TiN pH electrodes were made using RF magnetron spray producing 85 nm of TiN layer on the underlying 2 inch × 2 inch 0.5 mm thick alumina substrate. As previously reported [43], the parameters of the sputtered-deposition process used for the manufacture of small metal nitride films under controlled conditions to produce the most effective solid pH sensor are Ti target of 99.95% purity with a power of 350 W sputter at 10:2 Ar: N_2_ at 2 mTorr pressure at room temperature. The TiN film electrode was then further refined with Nafion layer, by incorporating a 5 µL spin of 5% Nafion (Sigma) solution onto the active electrode, which was then fired at 210 °C for one hour under a vacuum of <10 mTorr using rapid thermal annealing (RTA). The RF magnetron exploded the 85 nm thick Nafion modified TiN electrode, as shown in Figure 2 (left) and Figure 2 (right) of the actual electrode.

### 2.2. Nafion Deposition and Annealing

The Nafion was deposited on sputter-fabricated 85 nm TiN. A total of 5 µL of 5% Nafion (Sigma)was spin coated on the pH sensitive TiN electrode. The Korea Vacuum Tech KVR-4000 Rapid thermal annealing (RTA) was used to cure Nafion to obtain good adhesion.After spin coating the modified electrode was annealed at 150 °C for 25 min.

### 2.3. SEM Characterization

Scanning electron microscopy (SEM) was used to examine the morphology of modified electrodes. The instrument Hitachi SU3500 was used for analysis. Samples were cleaned using isopropyl alcohol and completely dried on a hotplate before being mounted with carbon tape to the specimen stub. Images were captured using secondary electron detection, and settings (beam power and magnification) are displayed on individual images, as discussed in Section 3.2.

### 2.4. Sensing Protocol

Real-time recording between sensory TiN electrodes was performed using a high impedance Agilent 34410A digital multimeter, as well as a double-aging Ag | AgCl | KCl reference electrode (Sigma-Aldrich, St. Louis, MI, USA) connected to (-ve terminal) multimeter. Except for the reaction time, all measurements were taken at intervals of 20 s. To increase the signal-to-audio sensor ratio, an improved unity benefit booster was employed. All measurements were carried out at 22 °C with magnetic stirring after the electrodes had been soaked in a pH 7 buffer for an hour before calibration. Each potential recording had 30 data points, which were averaged to create individual measurement (this avoided the rapid shift that typically occurs during the first 30 s of recording due to electrode equilibration). The sensitivity, E^0^, hysteresis, and drift of the sensors were then determined using these results. While electrode drift was determined using the slope of the line-of-best-fit for the data at pH 12 across the measurement period, hysteresis was calculated using the difference between successive measurements at pH 12 [43]. The amount of time needed to bring an electrode within 3 mV (or 0.05 pH) of the stable potential was called the electrode reaction time [44].

The Hanna instruments commercial pH 4.7 and 10 buffers were used to test the performance of the TiN sensor by pH looping of 7-4-7-10-7 and 7-10-7-4-7 for three minutes. Between each measurement, an air burst was used to clean the electrodes. Sensitivity, E^0^, and sensory hysteresis were calculated from data points, and error bars with a 95% confidence interval were recorded. Using pH 7 buffer readings, short-term flow rate was calculated during the experimental period. The erosion level was represented by the best equity line from this data, and the errors of this measure were calculated using the worst equity line. Each sensor’s response time was calculated by exposing the active TiN electrode to pH 4 or 10 for 3 min, then monitoring the potential change for 60 s after the sensor was exposed to pH 10. The reaction time is defined as the time required to obtain 1 mV of steady power.

### 2.5. Response Time, Sensitivity, and Stability

The response time was calculated as the time required for the electrode potential to increase by 90% from its stable value. Electrodes were submerged in DI water overnight to track the stability of electrode responsiveness over time. The drift rate (in mV/h) was calculated using the line-of-best-fit approach’s slope. The TiN/Nafion electrodes were subjected to a variety of pH buffer solutions to evaluate the hysteresis, or memory effect, of an electrode. Firstly, pH was altered from 1.1 to 4.1 and then from 7.0 to 10.0, i.e., acidic to basic, and then in the reverse direction from basic to acid. The electrode response was observed for three minutes after being dipped into a new buffer solution, and between each buffer the electrodes were washed with distilled water and dried with pressure gun.

## 3. Results and Discussions

### 3.1. Nafion Deposition and Annealing

A study by Kinlen et al. [33] showed that many electrodes acting electrically as metal oxides react to a type of redox in a test solution. This problem was solved by using a heat-treated Nafion layer to shield the IrO_2_ electrode from redox species, such as interference from ferri/ferrocyanide. Iodide and permanganate ions could still damage the IrO_2_ sensor, since the Nafion layer could not completely shield all the active redox species. Additionally, the Nafion film added to the IrO_2_ electrode significantly lengthened the sensor’s reaction time in neutral pH, acidic, and basic regions. This increase in reaction time can be explained by the presence of inaccessible sulfonic acid areas, which consequently leads to a lower level of hydration, due to higher hydrophobicity. These sulfonic sites have low acidity (6–9 pKa values) resulting in slow proton transfer rates. Lonsdale et al. [44] have reported an unprotected (excluding any layer of electrode conversion) active RuO_2_ electrode to exhibit up to 300 mV fluctuations when exposed to redox agents. The addition of a Ta_2_O_5_ layer and Nafion onto the RuO_2_ electrode did show improvement and reduced the shift due to some redox species. However, it still failed to prevent disruption from stronger redox agents, such as those in wine and citrus juices. Nevertheless, since previous research demonstrated that Nafion does display characteristics of a redox inhibitor, it is therefore worth investigating it to improve the previously reported 30 mV shift of the TiN sensor [45].

The TiN electrodes were prepared as explained in Section 2.1, and 5 L of Nafion was spin-coated on top of 85 nm of TiN, as shown in Figure 3.

After applying Nafion to the surface of the electrode, the electrode was treated with an annealing process to improve the adhesion of the film. Previous studies have shown that annealing of RuO_2_ at elevated temperatures results in increased crystallinity, lower electrical resistance and decreased capacitance [46]; however, the effect of annealing on pH sensitivity is not specifically reported in the literature. Therefore, the effect of post-deposition annealing temperature on the pH-sensing properties of TiN thin film sputtered with Nafion modification was investigated here. The effect of annealing temperature and time for Nafion was investigated and is reported in Table 1. The effects of annealing conditioning on the effectiveness of the solution-cast Nafion membranes were investigated experimentally. Annealing alters water distribution within films. Modified Nafion electrodes were subjected to a ramped thermal annealing (RTA) process that produced Nernstian sensitivity for 25 min at 150 °C in a vacuum (10 mTorr) (rating 5). Drying time and humidity affect the final resistance and, thus, the performance of the formed membrane and the retention of the membrane composition. The inclusion of thermal annealing appears to ‘normalize’ transport structures of various sizes and clusters of Nafion, reducing the number of modified transport structures between the thin and thick layers [46].

The addition of Nafion altered the sensitivity of the pH sensor of the TiN electrode, as shown in Table 2. The thicknesses of the Nafion film had a direct effect on sensory function, i.e., pH sensitivity, redox inhibitory capacity, and duration of the sensory response. The thickness of the Nafion protective layer was optimized considering the concentration (5%) and amount of Nafion. Both thick layers (50 µL) and thin (5 µL) layers of Nafion were investigated: the 50 µL layer, which was the maximum amount of Nafion required to block the 30 mV redox shift, while remaining sensitive to pH, and the 5 µL layer, which was the minimum amount of Nafion needed to prevent 30 mV/h redox switching in the short reaction time. The thicker Nafion layers (15 µL, 25 µL and 50 µL) demonstrated an increase in both hysteresis and drift, whereas the thin layer of Nafion (5 µL) showed improved sensitivity and hysteresis (Table 2) caused by slow proton transfer at high pH, which is in line with the findings of Kinlen et al. [33]. In this work, Nafion reduced the disturbance caused by ascorbic acid and potassium permanganate, KMnO_4_, with a thinner layer proving to be more protective than a much thicker and harder Nafion layer, which is consistent with the findings of Kinlen et al. [33]. This phenomenon can be linked to the properties of Nafion, which upon hydration generates wrongly terminated channels that are strongly connected with cations. However, bigger channels might permit a slower migration of non-cationic species [47].

As can be shown in Table 1, the sensor drift, hysteresis, and Nernstian sensitivity were all improved by the thin layer of Nafion. This indicates that a TiN electrode modified with a thin coating of Nafion might be more suitable for applications in particular sample matrices; with “intermediate” amounts of disruptive redox chemicals, the thicker layers of Nafion displayed considerable hysteresis and drift values. The reaction time of the Nafion modified sensor is discussed in Section 3.4.

### 3.2. SEM Analysis

Secondary electron images of the top surface morphology of the layers formed on TiN (85 nm) coatings with Nafion (5 µL) at different magnifications post-annealing are shown in Figure 4. The top surface view image of the Nafion layer on the TiN electrode shows a relatively uniform and smooth surface with only a few granular precipitates (Figure 4a,b) on the coating surface; however, as the magnification increased (Figure 4c,d) a more detailed morphology of the structure of the TiN with Nafion was visible, comprising many pillars, and in some areas, adjacent growths aggregated to form a clustered structure. Between the clusters, deep valleys can be observed in Figure 4c. According to the image in Figure 4d, the Nafion–TiN layer exhibits a rough sectional morphology. Cracks and defects can be observed in this image. This contrasts with what is observed for SEM images of pure TiN films, as shown by Nana Sun et al. [48], where the surface of the TiN film is a very smooth. Having the thicker Nafion film deposited on the TiN sensor significantly changes the sensor’s behavior, as evident from the sensitivity results discussed in Section 3.3, and this could potentially be linked to the morphology of the Nafion film [49].

Annealing was found to be important, as the voltage readings were not available on air-treated electrodes, indicating that the complete layer of encapsulation, which occurs in water, was not performed over TiN. When annealed, an opaque white substance (when the electrode is dry) was formed, which is consistent with the fact that Nafion undergoes rapid polymerization when exposed to H_2_O [50]. This showed that the annealing process caused Nafion to create a hollow structure that forms a proton transmission space.

### 3.3. Sensitivity Testing

Three categories of the TiN with Nafion film were developed and tested, namely:(i)Pristine TiN (no layer of Nafion modification)(ii)TiN + N1 (N1—one layer of 5 µL of Nafion)(iii)TiN + N2 (N2—two layers of 5 µL of Nafion)(iv)TiN + N3 (N3—three layers of 5 µL of Nafion)

The properties of the Nafion membrane are affected by several variables, such as the deposition method, concentration and volume of the applied solution, Nafion polymer properties (such as side chain length), solvent, deposition temperature, the number of applied layers (for multilayered films), and the time between layer depositions. Due to its simplicity and affordability, the drop-casting deposition process was chosen for this study, and a consistent Nafion volume and 5% Nafion concentration solution was applied. After the electrodes achieved stable sensitivity values, we compared the characteristics of electrodes containing TiN + N1, TiN + N2, and TiN + N3 layers of Nafion. After a month of conditioning in water, TiN + N1 electrodes were examined (Figure 5). The sensitivity response of all the produced electrodes were equivalent to the Nernstian response (58.4 mV/pH at 21 °C) and to a conventional glass electrode (58.8 mV/pH).

In addition to being comparable to a commercially available glass electrode, in terms of performance, the TiN + N1 and TiN + N2 electrodes revealed only minor differences in performance (Figure 5). An increase in hysteresis, a decrease in sensitivity, and a linear response were observed for TiN + N3 electrodes. As with previously fabricated unaltered TiN electrodes [43], the sensitivity of Nafion-covered electrodes was comparable to and nearly equal to the theoretical value (58.8 mV/pH). With more Nafion layers present, there was also a minor drop in sensitivity; however, except for TiN + N3, it remained rather close to the Nernstain sensitivity.

### 3.4. Response Time

When referring to electrochemical sensors, the response time is typically defined as the interval between the start of the measurement and the moment at which the sensory output reaches 90% of the measurement. The pH sensor response time was the amount of time needed after the sensor was submerged in the sample solutions for the sensor energy to reach 90% of the recorded voltage. The improved pH sensor was submerged in standard pH bath solutions with pH values of 4.33, 6.68, and 9.21 to measure the response time. Response times for the TiN sensor were discovered using an electrochemical field to record the sensor’s output. All tests were completed in triplicate and the average reaction time is shown in Table 3. What is significant here is that the response time of the TiN/Nafion pH sensor is now only approximately 12 s in neutral solutions. This is a much shorter reaction time than for acidic and alkaline solutions (see Table 3), and more importantly, these response times are considerably shorter than for other solid-state sensors. For the RuO_2_/Ta_2_O_5_/Nafion sensor, the response time is 136 s at neutral pH, and longer for acidic and alkaline solutions, which would be inappropriate for some applications. The TiN/Nafion sensor, having such a short response time in various pH solutions compared to the RuO_2_/Ta_2_O_5_/Nafion sensor, is therefore very suitable for widespread applications.

### 3.5. Sensor Stability

One of the most crucial indicators of a sensor is the stability of the pH sensor. All studies were carried out at room temperature to assess the stability of the TiN sensor (25 °C). The sensor was submerged in buffer solutions with pH standards of 2, 4, and 7. Each analysis took 10 min to complete, and an electro-chemical workstation logged the data every 30 s. The TiN pH sensor was cleaned in DI water and dried in a vacuum after each measurement. The outcomes are depicted in Figure 6 below:

In this study, relative statistical analysis was used. The analysis’s findings show that the observed output potential in pH 4 buffer solution had a standard deviation of 1.1 mV, compared to less than 1 mV for pH 7, and 2.1 mV for pH 2. As a result, throughout a 10-minutes period, the standard deviations in all three solutions were less than 2 mV. Additionally, a month-long study on the stability of the TiN pH sensor was conducted, with weekly measurements and records of the solutions’ pH levels and equilibrium potentials. Each measurement was carried out three times in a single day, and the mean linearity value was calculated as shown in Table 4 Linear fitting was used to determine the developed pH sensor’s sensitivities for each measurement and reproducibility of the sensor was achieved for each of these stability tests. The sensitivities on the first, seventh, fourteenth, twenty-first, and thirty-fifth days were 58.5 mV/pH, 55.8 mV/pH, 56.2 mV/pH, 56.5 mV/pH, and 56.2 mV/pH. Less than 2.5 mV/pH represented the highest variation of the sensitivity.

Numerous research papers have investigated solid-state pH sensors, especially with magnetron sputtering. For example, Uppuluri et al. [51] developed a pH sensor with RuO_2_ electrodes deposited on screen-printed electrodes with a sensitivity of 56.1 mV/pH, and a response time of 2 s in less than pH 7. Xu et al. The author of [52] reported a RuO_2_ pH sensor using RF sputtering, with a sensitivity of 54.5 mV/pH and potential drift of 2 mV/h, with a 20 s response time in the neutral region. Comparing the performance of the sensor developed here with the pH sensors reported by other scientists, the TiN/Nafion sensor displays better drift value of 0.93 mV/h (Table 2), and a stable Nernstian sensitivity (Table 4), but most importantly has a much shorter response time of 12 s in neutral solutions. Most importantly, there was no leaching of the modified membranes observed that can affect the sensitivity, which was an issue in previous modified pH sensors [52].

## 4. Conclusions

A solid-state TiN pH sensor was developed and tested. The TiN performance in redox samples was compared to a previously developed RuO_2_ sensor. The potential value of the TiN (E^0^) electrode varies significantly less (only 30 mV) in the reducing agents compared to the RuO_2_ sensor, which displayed a tenfold variation (300 mV). This means that for the first time, accurate pH readings can now be obtained from samples such as wine and fresh orange juice, previously not possible with other metal-oxide pH sensors. This can be achieved by simply blocking a possible 30 mV variation with only a small amount of Nafion. Previously, Nafion was used successfully to prevent redox transfer in the RuO_2_ sensor; however, a thick layer of Nafion was required to prevent the large 300 mV variation, and this in turn increased the response time of the sensor. However, with the TiN electrode, because only a small (30 mV) variation was required for blocking, a significantly thinner layer of Nafion was required. Experimental results showed that a 5 µL layer of Nafion not only was able to block the 30 mV variation, but also provided a significant reduction in reaction time, from 136 s to 17 s in a neutral pH, and in slightly acidic medium from 55 s to 20 s, and in basic regions from 38 s to 26 s. In addition, the TiN/Nafion sensor showed a much smaller drift (2 mV/h) than the RuO_2_/Ta_2_O_5_/Nafion electrode (7.2 mV/h). This makes the TiN/Nafion electrode the best candidate for various matrix applications without any conditioning or matrix matching protocols. Future work in this field would comprise of interference testing by using potential species that can alter pH sensitivity, such as ascorbic acid, uric acid etc. That study would involve implementing profilometry or AFM characterization to investigate the surface and its porosity, in order to better understand the key factors that affect the capability to block interfering species.

## Figures and Tables

**Figure 1 sensors-23-02331-f001:**
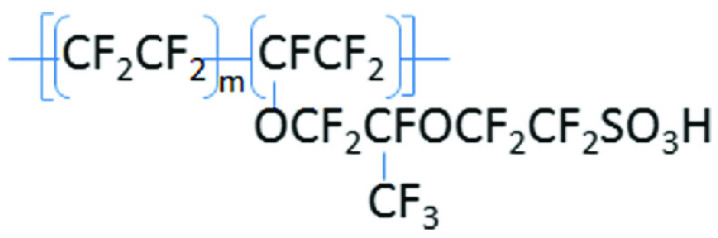
Chemical structure of Nafion [18].

**Figure 2 sensors-23-02331-f002:**
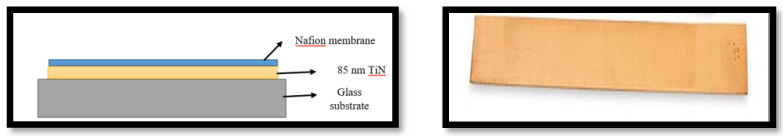
Schematic representation of 85 nm thick TiN electrode with Nafion layer (**left**) and an image of the actual Nafion modified electrode (**right**).

**Figure 3 sensors-23-02331-f003:**
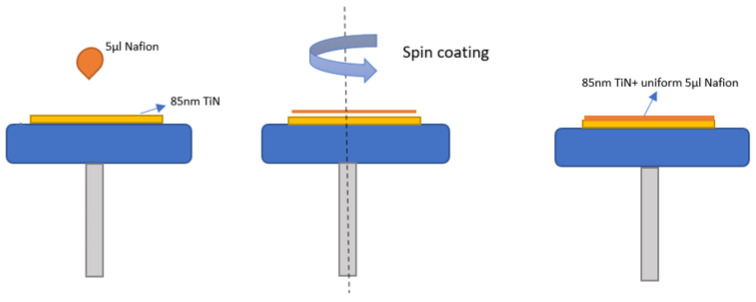
Schematic showing deposition of Nafion membrane on 85 nm TiN electrode using spin coating.

**Figure 4 sensors-23-02331-f004:**
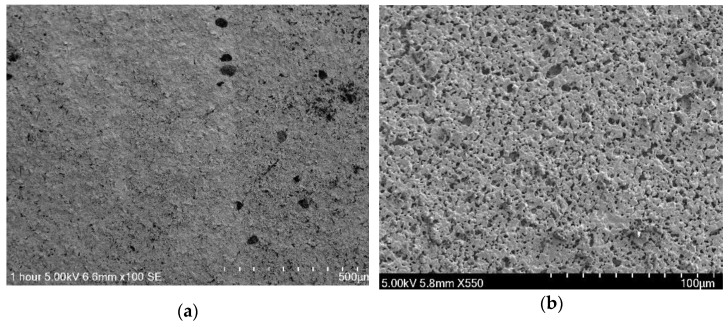
SEM secondary electron images taken with Hitachi SU3500 of (**a**–**d**) 85 nm TiN sensor modified with Nafion film at various magnifications, and (**e**) unmodified 85nm TiN sensor [45].

**Figure 5 sensors-23-02331-f005:**
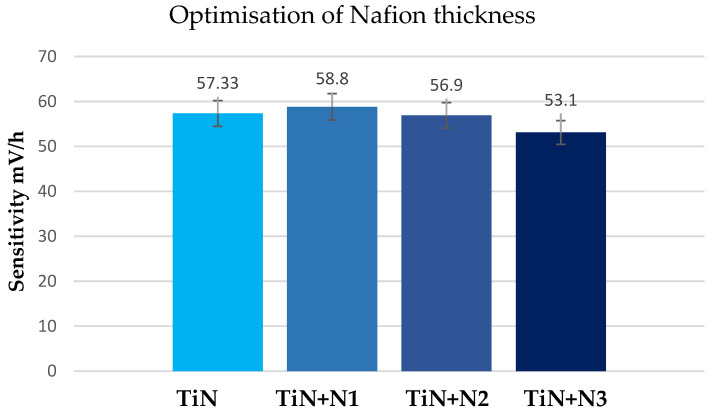
Thickness optimization of Nafion on 85 nm TiN film by assessing the sensitivity as a key performance indicator.

**Figure 6 sensors-23-02331-f006:**
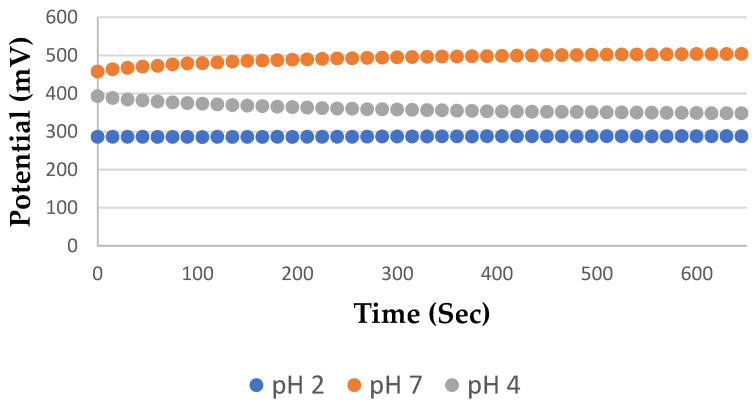
Graphical representation of potential difference vs. time for pH 2, 4 and 7.

**Table 1 sensors-23-02331-t001:** (**a**) Nafion annealing time and temperature optimization based on pH sensitivity ranking. (**b**) pH Sensitivity ranking criteria.

(a)				
**Time (min)**	**Temperature (°C)**			
	25	50	100	150
10	1	1	2	2
15	2	2	2	2
20	2	3	3	4
25	4	4	4	5
**(b)**				
**Ranking**		**Sensitivity (mV/pH)**		
1		30–35		
2		35–40		
3		40–45		
4		45–55		
5		55–58 (Close to Nernstian)		

**Table 2 sensors-23-02331-t002:** Nafion thickness investigation using optimized annealing time of 25 min and temperature of 150 °C as determined in Table 1.

Amount of Nafion (µL)	Sensitivity (mV/pH)	Hysteresis (mV)	Drift (mV/h)
50 (thick)	56 ± 0.92	56.1 ± 9.4	30.10
25	56.5 ± 0.88	34.5 ± 5.9	33.38
15	57.2 ± 1.7	10.7 ± 0.40	15.99
5 (thin)	58.5 ± 0.54	0.57 ± 0.29	0.920

**Table 3 sensors-23-02331-t003:** TiN pH sensor response time for different pH solutions.

pH	Response Time (s)
4.33	35
6.68	12
9.21	21

**Table 4 sensors-23-02331-t004:** Linearity and sensitivity recorded over a period of 35 days.

	Interval at Which Linearity Was Measured (Days)					
	1st	7th	14th	21st	28th	35th
**Linearity (R^2^)**	0.9999	0.9999	0.9998	0.9996	0.9991	0.9986
**Sensitivity** **(mV/pH)**	58.5	55.8	56.2	56.5	56.2	56.1

## Data Availability

The data presented in this study are available on request from the first author. The data are not publicly available as further study will be carried out using the same data.

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
