# Peer review of "Fabrication and Optimization of Nafion as a Protective Membrane for TiN-Based pH Sensors"

_sensors, 2023, doi:10.3390/s23042331_

Round 1

Reviewer 1 Report

In the manuscript entitled “Fabrication and optimisation of Nafion as a protective membrane for TiN based pH sensors”, S. P. Shylendra et al. have developed a solid-state modified pH sensor, consisting of active electrode consisting of TiN film with a protective membrane of Nafion, by the use of RF magnetron sputtering technology.

In the experimental section, the authors should add information about the buffer solutions used for the sensing protocol.

In my opinion the paragraph regarding “Nafion deposition and annealing” should be added in the experimental section.

Information about the analysis techniques should be added in the experimental section; moreover, in the experimental section, the authors should add the equations used to obtain the response time, and the sensor sensitivity.

The authors should indicate how they perform the measurement of Nafion thickness.

The authors should perform chemical analyses on pristine TiN substrate and on Nafion-modified TiN, also evaluating the effect of the thermal annealing on the chemical status of the membrane, since the sensing mechanism is affected by the chemistry at the interface.

The authors should evaluate and report the possible sensing mechanism of their developed pH sensor.

Which is the sensor reproducibility? Discuss.

In my opinion, to evaluate in a correct way the effect of the Nafion protective membrane, the authors should evaluate the morphology of the unmodified TiN electrode and those modified with different Nafion thickness and annealing times/temperatures.

The English style is acceptable.

I can accept this manuscript with major revisions.

Reviewer 2 Report

The first paragraph of the Introduction says very little about the 100 years development of the glass electrode and has with some errors: Glass electrodes are not necessarily large nor fragile. Emphically, they should not be stored in potassium chloride but in a buffer. Ref 5 appears to bear no relationship to the text. Also, calibration (buffering) is generally necessary before use; however, this appears also to be the case in the present study (Section 2.2).

Section 2 makes no mention of the dimensions of the TiN electrode nor of the means of connection to the reference electrode and measurement instrumentation. In 2.2, what does “highly effective” mean? Also, what is a “double-aging” reference electrode

Section 3 is much too long and written in a complicated, verbose style, often in a non-scientific manner. This makes the main advances reported, i.e. the redox insensitivity of the sensor to pH change, difficult to discern

It is surprising that there are no figures showing sensor response to pH change. Also, some data from practical applications would be expected

Round 2

Reviewer 1 Report

In my opinion, the revised version of the manuscript is acceptable for publication.

Reviewer 2 Report

The authors have made some alterations to the text as a result of my comments. I am not satisfied completely but am now content to recommend publication